# Genome-Wide Identification and Transcriptional Expression Profiles of *PP2C* in the Barley (*Hordeum vulgare* L.) Pan-Genome

**DOI:** 10.3390/genes13050834

**Published:** 2022-05-07

**Authors:** Xiao-Tong Wu, Zhu-Pei Xiong, Kun-Xiang Chen, Guo-Rong Zhao, Ke-Ru Feng, Xiu-Hua Li, Xi-Ran Li, Zhao Tian, Fu-Lin Huo, Meng-Xing Wang, Weining Song

**Affiliations:** 1State Key Laboratory of Crop Stress Biology for Arid Areas, College of Agronomy, Northwest A&F University, Yangling, Xianyang 712100, China; wu.xiaotong@nwafu.edu.cn (X.-T.W.); xiongzhupei@163.com (Z.-P.X.); chenkunxiang2022@163.com (K.-X.C.); zgr@nwafu.edu.cn (G.-R.Z.); fkr@nwafu.edu.cn (K.-R.F.); lxhfkr@nwafu.edu.cn (X.-H.L.); lixiran@nwafu.edu.cn (X.-R.L.); tian.zhao@nwafu.edu.cn (Z.T.); huoful@163.com (F.-L.H.); 2College of Agronomy, Jiangxi Agricultural University, Nanchang 330045, China

**Keywords:** PP2C gene family, barley pan-genome, positive selection, expression profiles

## Abstract

The gene family protein phosphatase 2C (PP2C) is related to developmental processes and stress responses in plants. Barley (*Hordeum vulgare* L.) is a popular cereal crop that is primarily utilized for human consumption and nutrition. However, there is little knowledge regarding the *PP2C* gene family in barley. In this study, a total of 1635 *PP2C* genes were identified in 20 barley pan-genome accessions. Then, chromosome localization, physical and chemical feature predictions and subcellular localization were systematically analyzed. One wild barley accession (B1K-04-12) and one cultivated barley (Morex) were chosen as representatives to further analyze and compare the differences in *HvPP2Cs* between wild and cultivated barley. Phylogenetic analysis showed that these *HvPP2Cs* were divided into 12 subgroups. Additionally, gene structure, conserved domain and motif, gene duplication event detection, interaction networks and gene expression profiles were analyzed in accessions Morex and B1K-04-12. In addition, qRT-PCR experiments in Morex indicated that seven *HvMorexPP2C* genes were involved in the response to aluminum and low pH stresses. Finally, a series of positively selected homologous genes were identified between wild accession B1K-04-12 and another 14 cultivated materials, indicating that these genes are important during barley domestication. This work provides a global overview of the putative physiological and biological functions of *PP2C* genes in barley. We provide a broad framework for understanding the domestication- and evolutionary-induced changes in *PP2C* genes between wild and cultivated barley.

## 1. Introduction

Phosphorylation and dephosphorylation are a pair of important protein modification processes that can influence multiple metabolic and biochemical reactions [1]. Depending on the primary sequence and catalytic mechanism, protein phosphatases (PPS) are divided into four major classes: phosphoprotein phosphatases (PPP), Mg^2+^- or Mn^2+^-dependent protein phosphatase (PPM)/protein phosphatase 2C (PP2C), phosphotyrosine phosphatase (PTP) and aspartate (Asp)-dependent enzyme families [1]. Previous studies showed that PP2C is related to developmental processes and stress responses in plants. In the nucleus, *AtPP2C5* acts as an *MAPK* phosphatase that is involved in seed germination, ABA-inducible gene expression and stomatal closure [2]. In Arabidopsis thaliana, *AP2C3* also mediates stomatal development by inducing the ectopic proliferation of epidermal cells [3]. In Komatsu’s study [4], *PpABI1B* single disruptants (ppabi1b) and *PpABI1* double disruptants (ppabi1a/b) and double disruptants (ppabi1a/b) were constructed. The ppabi1a/b of group A PP2C enhanced the tolerance of water-associated stresses by accumulating LEA proteins and soluble sugars at the maximum level [4]. By constructing transgenic *AtPP2CA*-antisense plants, Sari found that *AtPP2CA* is a negative regulator of ABA responses during cold acclimation [5].

Barley (*H.*
*vulgare*) is one of the world’s oldest cultivated crops, and is the fourth most abundant cereal in terms of both area and tonnage harvested (FAO: http://faosta.fao.org, accessed on 1 December 2021). Barley is mainly used for feeding and human consumption, such as in the brewing industry [6]. Compared with wheat, barley shows stronger environmental adaptability. Therefore, it can grow in harsh environments and maintain harvestable yields, allowing growers to avoid food crises. Barley is a diploid plant and has 14 chromosomes. In the Triticeae, barley is an excellent model for plant genetic research due to its abundant genetic diversity and small genome (5.1 G) [7].

It is generally accepted that barley domestication is a single event that happened in the Fertile Crescent [8]. Nevertheless, some other researchers consider Tibet or the Near East as secondary centers of origin [9,10,11]. During the domestication and improvement process, *Btr*, *Ppd*, *Vrs*/*int-c*, *nud*, *SD* and *Vrn*/*Sgh* have been regarded as the most critical genes or QTLs controlling barley shattering, photoperiod, row type, naked caryopsis, grain dormancy and vernalization. Among them, shattering has attracted widespread concern as a domestication gene because it is the only trait that directly distinguishes wild and cultivated barley. Non-brittle rachis 1 (*btr1*) and non-brittle rachis 2 (*btr2*) are two tightly linked genes located on barley chromosome 3H [12,13]. On the one hand, *Btr1* and *Btr2* are dominant alleles that generate wild barley with brittle rachises and eventual shattering [14,15]. On the other hand, the genotype of nonbrittle rachis cultivated barley is either *btr1Btr2* (*btr1*-type) or *Btr1btr2* (*btr2*-type), resulting in non-shattering spikes [16]. Another essential gene of domestication is *Ppd-H1*, which regulates flowering time and is located on the short arm of chromosome 2H [17,18]. According to James’ findings, while mutant alleles at the *Ppd-H1* and *Ppd-H1* photoperiod loci arose before domestication, mutant vernalization nonresponsive alleles at the *VRN-H1* and *VRN-H2* loci occurred after domestication [19,20]. However, the time and place of barley domestication are still unclear. Additionally, little is known about the mechanism of domestication. Thus, the study of domestication and evolutionary relationships might reveal information about barley cultivation expansion and subsequent growth, including the crop’s adaptability to new settings and reaction to human selection.

In the past decade, owing to the high degree of genomic variation, research has demonstrated that a single reference genome does not fully represent the diversity of a species [21]. The concept of pan-genomes was first proposed by Tettelin in 2005 [22]. Pan-genomes include multiple genetic resources and show more information about the entire species. Therefore, barley pan-genomes are valuable for research on genetic studies, barley domestication and improvement, adaptive evolution and plant breeding [22]. In recent years, some studies have been conducted to investigate genomic variations and functional genomics, particularly in rice [23] and soybeans [24]. The first barley pan-genomes involving 20 accessions were assembled in 2020, and included one wild barley and nineteen cultivated barley accessions [25]. The cultivated accessions include the reference cultivar Morex, two current or former elite malting varieties, two founder lines of Chinese barley breeding, two genotypes with high transformation efficiency, and a successful German variety [25].

Barley’s domestication process and adaptation to evolution have long been the focus of researchers all over the world. The recent sequencing of the barley pan-genome provides an excellent opportunity to accomplish these missions. In this sense, we identified HvPP2Cs in this barley pan-genome, reported their protein characteristics and chose two accessions, B1K-04-12 and Morex, to represent wild and cultivar materials, respectively. Then, we compared phylogenetics, and multiple structural and functional characterizations of HvPP2Cs between these two accessions. Finally, we analyzed the expression pattern of different tissues and treatments under different stresses. Our first aim in this study was to provide a global overview of the putative physiological and biological functions of PP2C genes in barley. We provide a broad framework for understanding domestication- and evolutionary-induced changes in *PP2C* genes in wild and cultivated barley. We also intend to provide insights into the genetic improvement of barley breeding strategies.

## 2. Material and Methods

### 2.1. Identification of PP2C Genes in H. vulgare

To identify *PP2C* genes in *H. vulgare*, the barley pan-genome sequences and their corresponding annotation information were downloaded from https://webblast.ipk-gatersleben.de/downloads/barley_pangenome/, accessed on 1 December 2021 [25]. Then, the known PP2Cs in Arabidopsis thaliana were obtained from the EKPD 1.1 database (http://ekpd.biocuckoo.org/advance.php, accessed on 1 December 2021) [26]. Afterwards, a HMMER (biosequence analysis using profile hidden Markov models) [27] search was conducted on 20 barley pan-genome protein sequences using pfam [28] Stockholm seeds PF00481 and PF13672 [29]. Meanwhile, a blastp [30] procedure was produced with default parameters using 82 *AtPP2Cs* as a query against these pan-genome protein sequences. The results of the HMMER search and blastp were integrated using a Perl [31] script. Furthermore, conserved domains were predicted using the NCBI CDD/SPARCLE database [32]. Then, the sequences that lacked complete domains belonging to PP2C or PP2C super families were removed. Additionally, the remaining sequences were used as candidates for further analysis.

### 2.2. Prediction of Protein Physical and Chemical Parameters and Subcellular Localization

The physical and chemical parameters of identified protein sequences were calculated using the Expasy ProtParam tool [33], including molecular weight (MW), theoretical isoelectric point (pI) and amino acid composition. WoLF PSORT online service [34] predicted protein subcellular localization.

### 2.3. Alignment and Phylogenetic Analysis

Multiple alignments of peptide sequences were carried out using the MEGA version X [35,36] built-in ClustalW method. Then, to find the best protein model for the maximum likelihood (ML) tree, a model test was run in MEGA-X with default parameters. A maximum likelihood tree was then constructed using the best model found in MEGA-CC 10.0.4 [37] with Morex, B1K-04-12 and Arabidopsis thaliana PP2C protein sequences. The topology of the phylogenetic tree was used to classify HvPP2Cs into subgroups, referring to the previous studies on Arabidopsis thaliana and Oryza sativa [38], paper mulberry [39], woodland strawberry [40] and potato [41]. Then, all candidate sequences were renamed based on their subgroup classifications.

### 2.4. Analysis of Gene Structure, Chromosomal Locations and Conserved Motifs

Conserved motifs were predicted using MEME Suite Program Version 5.4.1 [42] for all *HvPP2C* sequences. The gene structure and chromosomal location information was extracted from the genome annotation gff3 files. Graphics were displayed using TBtools [43].

### 2.5. Analysis of Synteny and Detection of Duplication Events

The synteny analysis between the Arabidopsis genome, Morex genome and B1K-04-12 genome was investigated in TBtools using the plugin MCScanX method. The images of synteny analysis and duplication events were then visualized using TBtools software [43].

### 2.6. Calculation of Ka/Ks Values

The synonymous (Ks) and nonsynonymous (Ka) substitution rates of homologue genes in Morex and B1K-04-12 were calculated using TBtools software [43].

### 2.7. Analysis of Cis-Acting Elements in the Promoter Regions

The 2000 bp upstream genomic sequences of identified *HvPP2C* genes were extracted from each genome of 20 barley pan-genomes. Then, the upstream sequences were delivered to the PlantCARE database (http://bioinformatics.psb.ugent.be/webtools/plantcare/html/, accessed on 1 December 2021) to identify cis-acting elements in the promoter regions.

### 2.8. Network Interaction Analysis

The interaction network in which the HvPP2Cs are involved was investigated using the ExPASy String tool [44]. Then, GO and KEGG annotation and enrichment analysis were implemented using eggNOG-mapper v2 [45,46] and visualized in TBtools [43].

### 2.9. Transcriptional Analysis

A total of 524 barley RNA-seq data accessions (Appendix A) from 13 independent bioprojects (PRJNA294716, PRJNA382490, PRJNA400519, PRJNA428086, PRJNA431836, PRJNA471777, PRJNA541021, PRJNA578897, PRJNA658250, PRJNA668924, PRJNA681455, PRJNA704034 and PRJNA749617) [47,48,49,50,51,52] were obtained from the NCBI Sequence Read Archive (SRA) database. Then, Hisat2 [53] and Stringtie [54,55] were used to analyze the gene expression level. Additionally, the FPKM value (fragments per kilobase of transcript per million fragments mapped) was calculated for each *PP2C* gene. Morex was used as the reference genome. To obtain a more distinct heatmap, log10-transformed (FPKM+1) values were used to display the results.

### 2.10. Plant Materials and Growth Conditions

Barley seeds (*H. vulgare* L. cv. Morex) were surface-sterilized then soaked in distilled water in a greenhouse. The low-pH and aluminum treatments were performed in a hydroponic environment. Briefly, seeds of barley were surface-sterilized in 5% sodium hypochlorite and incubated in the dark at 4 °C for stratification. Then, the seeds were placed on Petri dishes filled with moist filter paper and placed in a growth chamber at 25 °C in the dark. After 48 h, the germinated seeds were transferred to 21 hydroponic containers containing 2.5 L of Hoagland’s nutrient solution for 7 days, and the medium was changed every 24 h to maintain their composition. After 7 days, seedlings were cultured at pH = 6.0, pH = 4.0 and pH = 4.0 with 10 μM of bioavailable Al^3+^ ions for 1 or 7 days, respectively. A maximum of 12 seedlings were placed in one container, which was considered as one replicate, and each experimental combination was set up as three replicates.

### 2.11. RNA Isolation and qRT-PCR Analysis

Total RNA was extracted from the roots and shoots of each treated 14-day-old barley seedling using the TIANGEN RNA simple Total RNA Extraction kit (DP419). The quality of total RNA was detected by 1% agarose gel electrophoresis. The concentration of RNA was detected using a NanoDrop-2000 UV spectrophotometer. Samples that had an A260/A280 between 1.8 and 2.0 were used for reverse transcription, and then reverse transcribed into cDNA using an Evo M-MLV reverse transcription kit (Accurate Biotechnology (Hunan) Co., Ltd., Hunan, China). The reverse transcription assay was based on a 20 µL reaction volume containing 1 µg of total RNA. A SYBR Green Premix Pro Taq HS qPCR Kit (Accurate Biotechnology (Hunan) Co., Ltd., Hunan, China) was used for the qPCR experiment. The qPCR assay was based on a 20 µL reaction volume containing 10 µL of 2X SYBR Green Pro Taq HS Premix (ROX plus), 4 µL of each forward and reverse primer and 2 µL of cDNA. The amplification reactions were performed using the QuantStudio 7 Flex Real-time Fluorescent Quantitative PCR System with an initial denaturation at 95 °C for 30 s, followed by 40 cycles of amplification at 95 °C for 5 s and 58 °C for 34 s. Melting curves were acquired using the following procedure: 95 °C for 15 s, 58 °C for 60 s and 95 °C for 15 s. The used primers are listed in Appendix A. One-way ANOVA and Waller–Duncan multiple comparisons were performed between treatments using SPSS22 [56]. The results were visualized in R Studio [57].

## 3. Results

### 3.1. Genome-Wide Identification of PP2C Genes in the Barley Pan-Genome

A total of 1635 candidate *HvPP2C* genes were identified from 20 barley pan-genome accessions. The number of each accession is shown in Table 1. Both an HMMER search and the Blastp program were used for identification. The sequences were removed for further study, which revealed that they lacked complete conserved domains belonging to PP2C super families. As a result, Hockett had the most members with 88 *HvPP2Cs*, followed by Morex with 85 HvPP2Cs. Among all cultivated barley, Igri had the fewest members with 79 *HvPP2Cs*, followed by Akashinriki, Barke, Golden_Promise and HOR_3365, which have 80 *HvPP2Cs*. There was just one wild accession, B1K-04-12, among the 20 pan-genome accessions, and it had the smallest *HvPP2C* group, with only 79 members. The number of *HvPP2C* genes in each accession is similar to that of Arabidopsis thaliana, which contains a total of 82 sequences (EKPD: http://ekpd.biocuckoo.org/, accessed on 1 December 2021).

### 3.2. Physical and Chemical Feature Prediction and Subcellular Localization of HvPP2Cs

The physical and chemical features of all candidate genes were predicted by Expasy online services, and the results are shown in Appendix A. The lengths of the candidate genes varied from 116 amino acids to 1744 amino acids in Hockett. Each accession had a different average length, ranging from 403 amino acids in HOR 3081 to 429 amino acids in Hockett. Furthermore, the theoretical isoelectric point (pI) and average molecular weight (MW) ranged from 4.35 to 11.64 kDa and 12,769.69 to 191,804.35, respectively.

A summary of the subcellular localization of the two chosen accessions in this study is shown in Table 2. The results show that the PP2Cs are randomly distributed in the barley cells. Most of the PP2C genes are located in the chloroplast, and some of them are also located in the cytosol and nucleus. Only a few of them are located in other cellular components such as the cytoskeleton, peroxisome, endoplasmic reticulum and vacuolar membrane. Detailed information on the subcellular localization of each gene in the 20 barley pan-genome accessions is shown in Appendix A.

### 3.3. Phylogenetic Analysis

A phylogenetic ML-tree was constructed with all 1635 HvPP2C sequences (Appendix A). In order to obtain a clear view of the evolutionary relationship, two accessions, Morex and B1K-04-12, were chosen to represent cultivar and wild materials, respectively. Then, another ML-tree was constructed between 82 known Arabidopsis thaliana PP2C sequences, 85 identified barley Morex and 78 B1K-04-12 sequences, using MEGA-X software (Figure 1). The HvPP2Cs were divided into 12 subgroups based on the phylogenetic results, which is consistent with previous studies in Arabidopsis thaliana and Oryza sativa [38], paper mulberry (*Broussonetia papyrifera* L.) [39], woodland strawberry (*Fragaria vesca* L.) [40] and potato (*Solanum tuberosum* L.) [41]. As a result, subgroup A possessed the most members (13 in B1K-04-12 and 14 in Morex), followed by subgroup F (11 in B1K-04-12 and 11 in Morex). Additionally, subgroups J and U, which were identified in Arabidopsis thaliana, did not exist for all barley PP2C members in this study. Compared with B1K-04-12, cultivar Morex has five more members in subgroup K than wild accession B1K-04-12. More research is needed to determine whether these variations in subgroup K were driven by domestication or simply individual differentiations. The code names “Morex” and “FT11”, which were used in the barley pan-genome research (25) of accessions Morex and B1K-04-12, remained in the renaming system in this study. Appendix A shows the MEGA-X model test result of this ML-tree. Additionally, Appendix A shows the subgroup classification and renaming information of the Morex and B1K-04-12 PP2C genes.

### 3.4. Protein Motifs and Conserved Domain Analysis

Two phylogenetic trees were constructed using B1K-01-12 (Figure 2a) and Morex (Figure 3a) PP2C protein sequences, which represent wild and cultivated barley, respectively. The phylogenetic topology results were used to arrange the order for the protein motif, conserved domain and gene structure analysis.

The MEME motif search tool was used to identify common motifs. Ten conserved motifs were identified both in Morex and B1K-04-12. In wild barley B1K-01-12, seven motifs (1, 3, 4, 5, 6, 7 and 9) were present in the majority of the sequences, whereas other motifs (2, 8 and 10) were specific to the remaining sequences (Figure 2b). Meanwhile, five motifs (1, 2, 3, 6 and 7) were distributed in the majority of the sequences in cultivated barley Morex (Figure 3b). The width, sites and E-value of Morex HvPP2C proteins and their conserved motifs are shown in Appendix A.

The conserved domain was predicted using the NCBI CDD/SPARCLE database (Figure 2c and Figure 3c). All HvPP2C sequences have a conserved domain that belongs to the PP2C family. Most of them have a PP2Cc domain, with the exception of one in Morex (Horvu_MOREX_7H01G494000.1) and one in B1K-04-12 (Horvu_FT11_7H01G466900.1). These two genes have a PLN03145 domain belonging to the PP2C super family. Additionally, domains CAP_ED, PKc_like, MSCRAMM_ClfB and PHA03307 also appear in Morex and B1K-04-12 PP2C protein sequences. In particular, there are five unique domains that only appear in cultivated barley: NB-ARC, PLN03200, Arm, LRR and Rx_N. According to the annotation in the NCBI CDD database, the NB-ARC domain participates in signaling transduction, which regulates plant resistance. The PLN03200 domain is related to cellulose synthase-associated protein. Additionally, the Arm domain has 40 Armadillo/β-catenin-like repeats. These resistance-related domains occurred in the cultivar, suggesting that they might be the result of human breeding.

### 3.5. Gene Structure and Chromosome Location Analysis

The gene structures of B1K-04-12 and Morex *PP2C* genes were analyzed (Figure 2d and Figure 3d). In B1K-04-12 and Morex, the number of exons ranged from one to fifteen, with an average of five. Five genes in accession B1K-04-12 have just one exon and no intron, whereas ten genes in Morex have only one exon and no intron. As a result, the genes in the same group share a similar number of exons.

The chromosome locations (Figure 4) were extracted from the genome annotation gff file and displayed in the TBtools software. All 79 *HvPP2Cs* of B1K-04-12 and 85 *HvPP2Cs* of Morex were distributed on barley chromosome 1H to 7H, except one, *HvMorexPP2C14*, which is distributed on chromosome chrUn. As for the distribution location of a single chromosome, *HvPP2Cs* were largely distributed at both ends of chromosome 1H, 2H, 4H, 5H and 7H in both B1K-04-12 and Morex. The distribution of *HvPP2C* on chromosomes 3H and 6H, however, was relatively uniform.

### 3.6. Syntenic Relationships and Ka/Ks Analysis

The syntenic relationships of *PP2C* genes between Arabidopsis thaliana, wild barley B1K-04-12 and cultivated barley are displayed in Figure 5. The results revealed that the ortholog *PP2C* gene pairs (red and green lines in Figure 5) are mainly located on Arabidopsis thaliana’s chr1 and chr5 and barley’s chr4H, chr5H and chr6H chromosomes. The remaining sequences between *HvPP2Cs* and *AtPP2Cs* show little homology. This indicates that these genes might go through relatively large changes after speciation, preventing them from passing the ortholog identification test. As for the syntenic relationship within barley species between Morex and B1K-04-12 (gray lines in Figure 5), a total of 55 gene pairs were identified as syntenic. They are distributed on seven chromosomes of barley. The synonymous (Ks) and nonsynonymous (Ka) substitution rates of these 55 gene pairs were calculated with TBtools. The meaningless value of Ka/Ks was removed; the results are shown in Table 3.

Among these 55 gene pairs, 25 pairs had a meaningful result, including 24 purified selected gene pairs (with the Ka/Ks value << 1) and one positive selected gene pair (with the Ka/Ks value >> 1). This positive selected gene pair is Horvu_FT11_4H01G404600.1 and Horvu_MOREX_4H01G412600.1 and is renamed as HvFT11PP2C70 and HvMorexPP2C74, which belong to subgroup K and are located on barley chromosome 4H. These two genes both contain a PP2Cc domain and a MSCRAMM_ClfB domain.

### 3.7. Analysis of Cis-Acting Elements

Cis-acting elements were predicted using *HvPP2C* upstream 2000 bp sequences in Morex (Figure 6a) and B1K-04-12 (Figure 6b). The summary of predicted cis-elements is shown in Appendix A. A total of 3987 cis-elements were discovered in B1K-04-12 (1879) and Morex (2106). Among them, light-responsive elements are the most common, with a total of 1443, accounting for 36% of the total, followed by MeJA- (17%), abscisic acid- (12%), anaerobic induction- (6%), drought induction- (4%) and gibberellin- (4%) responsive elements. These six types of cis-elements account for 80% of the total.

### 3.8. Network Interaction, GO and KEGG Annotation and Enrichment Analysis

Network interaction analysis was performed using B1K-04-12 (Figure 7a) and Morex (Figure 7b) protein sequences, respectively. A total of 14 genes interacted with each other in B1K-04-12 and Morex.

The GO annotation information is shown in Appendix A. The enrichment results are shown in Figure 8. The KEGG annotation information is shown in Appendix A. The enrichment results are shown in Figure 9. The GO enrichment results suggest that HvPP2Cs are primarily involved in phosphoric molecular activities and interact with numerous chemicals, such as ester hydrolase, protein serine/threonine and phosphoproteins in biological processes. The KEGG enrichment results suggest that HvPP2Cs are involved in metabolism, protein phosphatases and associated proteins, the MAPK signaling pathway, plant hormone signal transduction, ribosome biogenesis signal transduction and environmental information processing.

### 3.9. RNA-Seq Analysis of HvPP2C Genes

RNA-Seq data were used to study the expression profiles of barley in 13 published bioprojects. The variety Morex was used as the reference genome. The gene expression level of *HvPP2C* genes was then normalized using the FPKM (fragments per kilobase million) values. In general, 63 *HvPP2C* genes were expressed in at least one tissue, stage or experimental treatment.

*HvPP2Cs* are related to developmental processes. A total of 32 *HvPP2Cs* were expressed at least in one stage during the early stages of barley development (Figure 10, bioproject PRJNA428086). Among them, two genes (HvMorexPP2C41 and HvMorexPP2C67) exhibited specific expression on day 0 of barley growth, eight genes on day 2 and seven genes on day 5. In addition, another RNA-seq result (Appendix A, bioproject PRJNA668924) showed that HvPP2Cs have different expression levels during inflorescence development and may be required for inflorescence indeterminacy and spikelet determinacy.

HvPP2Cs are also involved in the response to multiple biotic and abiotic stresses. The RNA-seq results demonstrate that multiple HvPP22C genes showed different expression patterns from wildtype to NPR1 overexpressed type and NPR1 knockdown type during their NPR1-mediated acquired resistance (AR) triggered by Pseudomonas syringae pv. tomato DC3000 (Appendix A, bioproject PRJNA431836). In Gao’s study (2018) [51], these ARs are potential resources to improve wheat resistance to *Puccinia triticina* L. HvPP2C genes also show different expression levels in abiotic stresses, such as high ambient temperatures (Appendix A, PRJNA658250), heavy metal stress (Appendix A, PRJNA382490), aluminum, low pH stresses (Appendix A, PRJNA704034) and salt stress (Appendix A, PRJNA578897).

### 3.10. Expression Pattern of HvPP2Cs in Aluminum and Low pH Stresses

Summarizing the results of all 13 RNA-seq analyses, the most abundant gene expression differences under experimental treatments were in aluminum and low pH, while many HvPP2Cs also showed tissue specific expression. Seven genes were randomly selected for qRT-PCR experiments from different expressed genes in aluminum and low pH experiment treatments or with tissue-specific expression summarized in other experiments.

To verify the effects of *HvPP2Cs* during barley stress responses, short- (1 day) and long-term (7 days) aluminum and low pH stresses were taken, and qRT-PCR was performed on barley tissue roots and shoots, respectively (Figure 11). In the shoot, the genes *HvMorexPP2C50*, *HvMorexPP2C68* and *HvMorexPP2C72* showed e upregulated expression of pH = 4.0, pH = 4.0 with Al^3+^ ions and pH = 6.0 short-term treatment. Additionally, *HvMorexPP2C46*, *HvMorexPP2C69* and *HvMorexPP2C74* showed upregulated expression of pH = 4.0 with Al^3+^ ions, and pH = 6.0 short-term treatment. In roots, *HvMorexPP2C46*, *HvMorexPP2C69* and *HvMorexPP2C72* showed upregulated expression of pH = 4.0, and pH = 4.0 with Al^3+^ ions, and *HvMorexPP2C74* also showed upregulated expression of pH = 4.0, and pH = 6.0 in short-term treatment. As for long-term treatment, *HvMorexPP2C46*, *HvMorexPP2C69* and *HvMorexPP2C72* showed upregulated expression at pH = 4.0, and pH = 4.0 with Al^3+^ ions, and *HvMorexPP2C69*, *HvMorexPP2C72* and *HvMorexPP2C74* showed upregulated expression under the pH = 6.0 condition. Among all the treatments, only two genes (*HvMorexPP2C59* and *HvMorexPP2C69*) showed downregulated expression after short-term pH = 6.0 treatment in shoots. Additionally, only one gene (*HvMorexPP2C59*) showed downregulated expression after a short-term pH = 6.0 treatment and a long-term pH = 4.0 with Al^3+^ ion treatment. The expression pattern of *HvPP2Cs* in aluminum and low pH stresses suggested that *HvPP2Cs* are sensitive to the stimulation of the external environment, thus confirming that these genes are involved in stress responses in barley.

## 4. Discussion

To better understand the role of *HvPP2Cs* in barley domestication and adaptive evolution, the differences in 20 barley pan-genome accessions were compared and summarized between wild accession B1K-04-12 and other cultivated varieties, taking Morex as a representative. First, in terms of gene quantity, wild accession B1K-04-12 has the least number of PP2Cs; only 78 being identified, and other cultivated materials have an average of 82. Second, in terms of distribution, most of these extra sequences of cultivated materials are located on the 7H chromosome and distributed in the cytosol, belonging to the K subfamily. Third, in terms of conserved domains and gene functions, NB-ARC, PLN03200, Arm, LRR and Rx N are five distinct domains that are solely found in cultivated barley. According to the NCBI CDD database annotation, the NB-ARC domain is involved in signaling transduction, which regulates plant resistance, and the PLN03200 domain is related to cellulose synthase-associated protein. Fourth, in terms of selective forces acting on the protein, when calculating the Ka/Ks value of homologous *HvPP2C* gene pairs of wild and cultivated materials, most homologous gene pairs undergo purification selection (Ka/Ks << 1), only a small proportion experience neutral selection, and a few homologous gene pairs experience positive selection. This suggests that *PP2C* is highly conserved in barley evolution. This phenomenon is extreme in the *PP2C* homologous gene pairs of Morex and B1K-04-12, in which only one pair of 25 pairs of effective Ka/Ks values is positively selected, and the remaining 24 pairs are negatively selected, with no neutral selection. Among seven genes used for the qRT-PCR, three of them had a meaningful Ka/Ks value. These include two purified selection genes, HvMorexPP2C50 and HvMorexPP2C72, with Ka/Ks values of 0.1718 and 0.3184, respectively, and one positive selection gene, HvMorexPP2C74, with a Ka/K values of 1.3583. To investigate whether this pair of positively selected genes was genuine, the homologues of the gene Horvu_FT11_4H01G404600.1 in the remaining 19 cultivars were identified. For the gene set composed of these 20 genes, pairwise Ka/Ks values were calculated. Out of a total of 190 results, there were 98 valid Ka/Ks results (Appendix A). Among them, 84 results were neutral selection or purification selection, and only 14 results were positive selection gene pairs. All of the positive selection gene pairs were the result of comparing the Horvu_FT11_4H01G404600.1 gene in the wild material B1K-04-12 with the homologous gene pairs in other cultivated materials. Therefore, it can be shown that this gene is indeed involved in the domestication process of barley and is subjected to strong selective forces. qRT-PCR confirmed this positive selection gene HvMorexPP2C74, showing the upregulated expression of pH = 6.0 in both shoot and root after short- or long-term treatments. It also showed the upregulated expression of pH = 4.0 with Al^3+^ ions in shoot after short-term treatment. This indicates that this gene might be involved in stress responses in barley. In addition to the PP2Cc domain, this gene also contains a MSCRAMM_ClfB domain. There is no published literature on the structure and function of this gene. Therefore, future in-depth studies on the differences between wild and cultivated varieties of this gene will be a good entry point for studying barley domestication.

## 5. Conclusions

In this study, a total of 1635 *HvPP2C* genes were identified in 20 barley pan-genome accessions. Two accessions were chosen to represent wild (B1K-04-12) and cultivated (Morex) materials, respectively. Subgroup classification, multiple alignment and phylogenetic analysis of known *AtPP2Cs* and newly identified *HvMorexPP2Cs* and *HvFT11PP2Cs* were carried out. Then, the gene and protein features were analyzed. The results indicate that the *HvPP2Cs* were conserved during evolution. The expression profile shows that seven *HvPP2C* genes (*HvMorexPP2C46*, *HvMorexPP2C50*, *HvMorexPP2C59*, *HvMorexPP2C68*, *HvMorexPP2C69*, *HvMorexPP2C72* and *HvMorexPP2C74*) are involved in the response to aluminum and low pH stresses. Fourteen positively selected homologous gene pairs were identified between wild accession B1K-04-12 and 14 other cultivars, indicating that these genes are important members of barley domestication. Our findings provide new insights into the functional evolution of *HvPP2Cs* during barley domestication.

## Figures and Tables

**Figure 1 genes-13-00834-f001:**
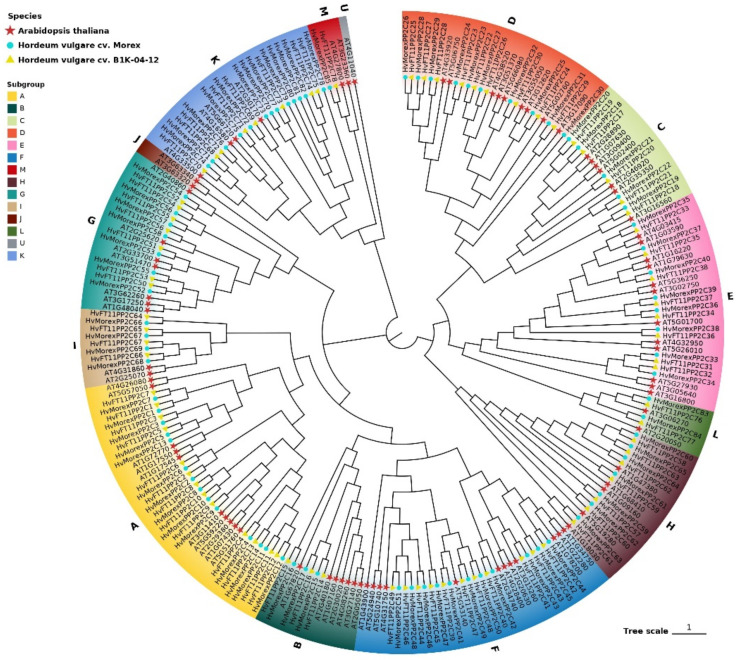
An ML-tree was constructed between 82 known Arabidopsis thaliana PP2C sequences (marked with red stars), 85 new identified barley Morex (marked with blue circles) and 78 B1K-04-12 sequences (marked with yellow triangles) using MEGA-X software. The background colors represent HvPP2C subgroups, which are labelled in the out layer.

**Figure 2 genes-13-00834-f002:**
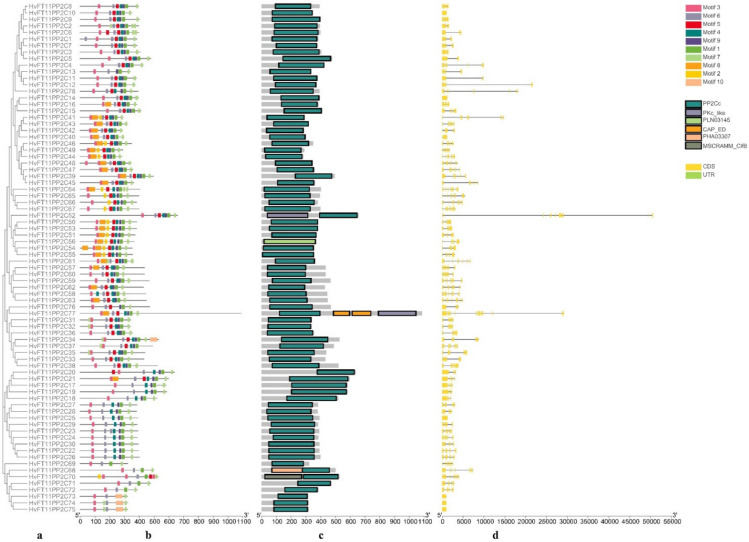
Phylogenetic, motif, conserved domain and gene structure analysis of HvPP2Cs in wild barley B1K-04-12. (**a**) Phylogenetic tree, (**b**) conserved motif distribution, (**c**) conserved domains and (**d**) gene structure.

**Figure 3 genes-13-00834-f003:**
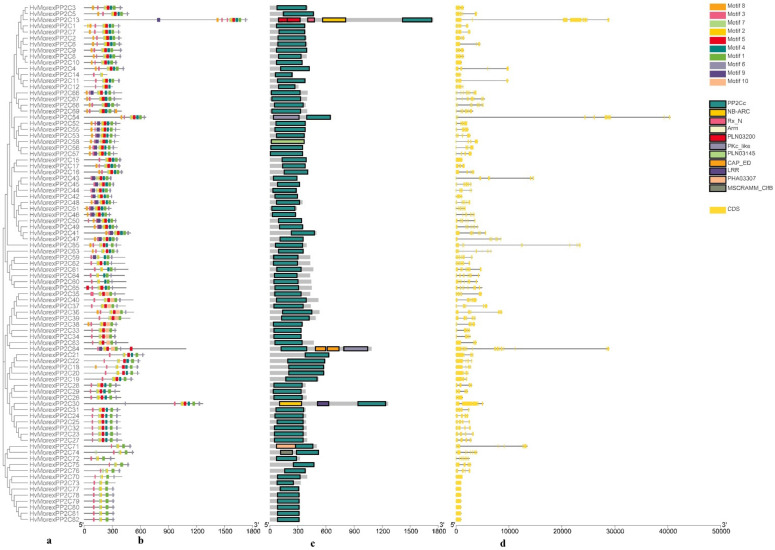
Phylogenetic, motif, conserved domain and gene structure analysis of HvPP2Cs in cultivated barley Morex. (**a**) Phylogenetic tree, (**b**) conserved motif distribution, (**c**) conserved domains and (**d**) gene structure.

**Figure 4 genes-13-00834-f004:**
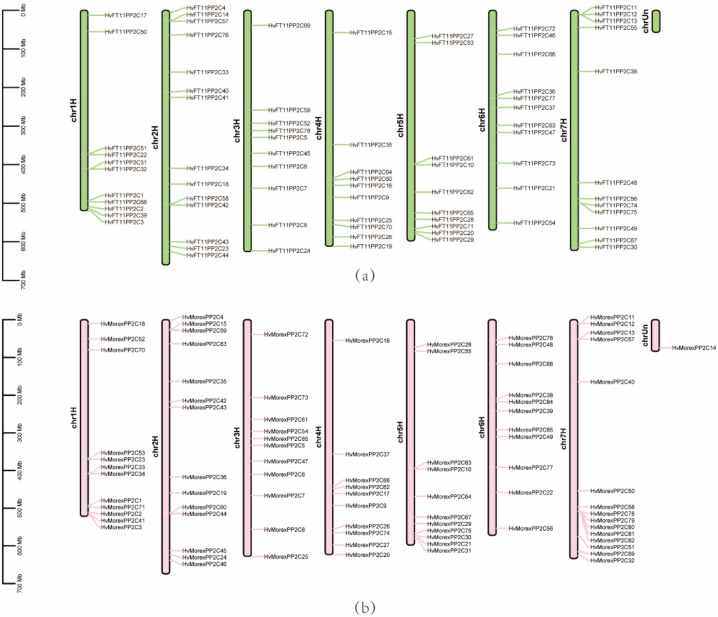
Chromosome distribution of *HvPP2Cs* in (**a**) B1K-04-12 and (**b**) Morex.

**Figure 5 genes-13-00834-f005:**
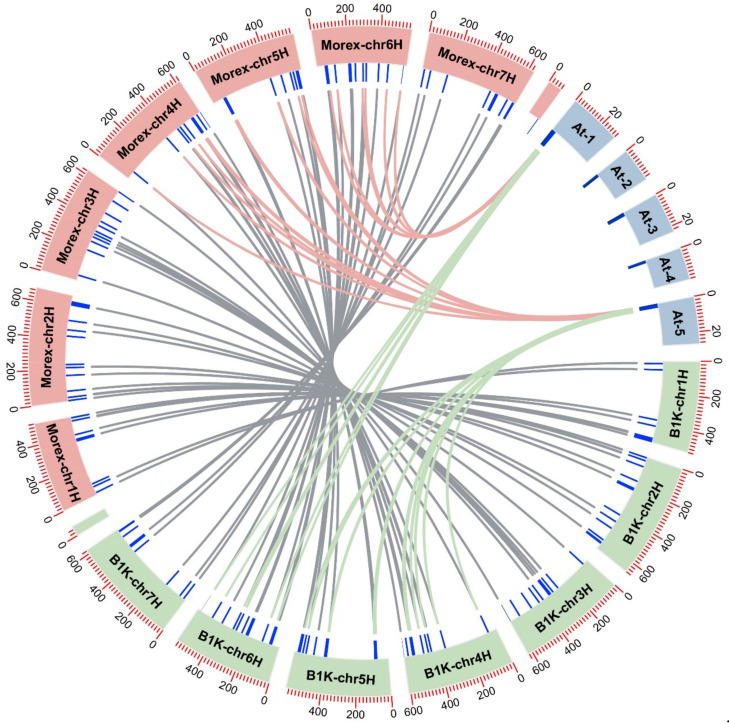
Synteny analysis of *PP2C* genes between 82 known Arabidopsis thaliana *PP2C* sequences (located on blue chromosomes), 85 newly identified barley Morex (located on red chromosomes) and 78 B1K-04-12 sequences (located on green chromosomes). The orthologous gene pairs between Arabidopsis and Morex are marked with red lines. The orthologous gene pairs between Arabidopsis and B1K-04-12 are marked with green lines. Additionally, the orthologous gene pairs between Morex and B1K-04-12 are marked with gray lines.

**Figure 6 genes-13-00834-f006:**
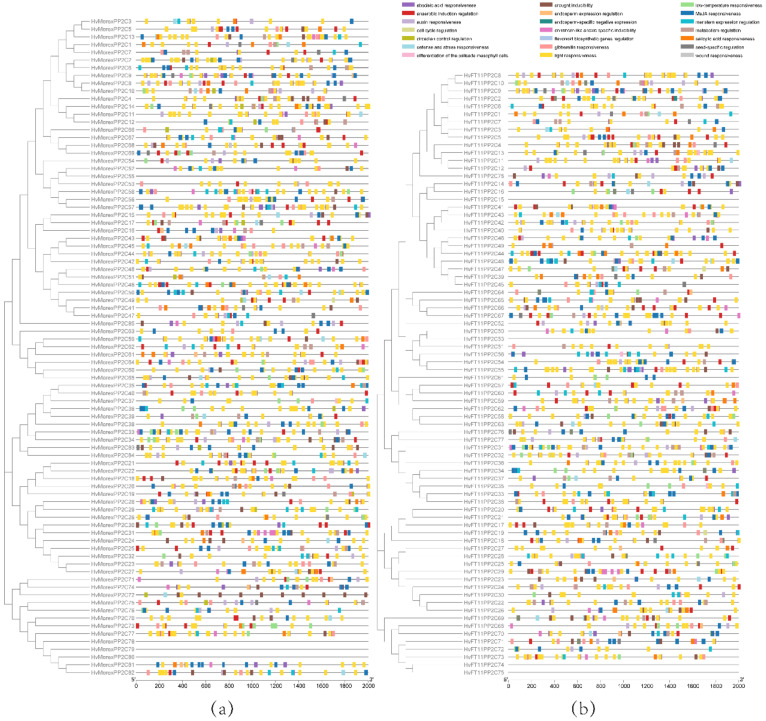
Cis-element analysis of HvPP2Cs in (**a**) Morex and (**b**) B1K-04-12.

**Figure 7 genes-13-00834-f007:**
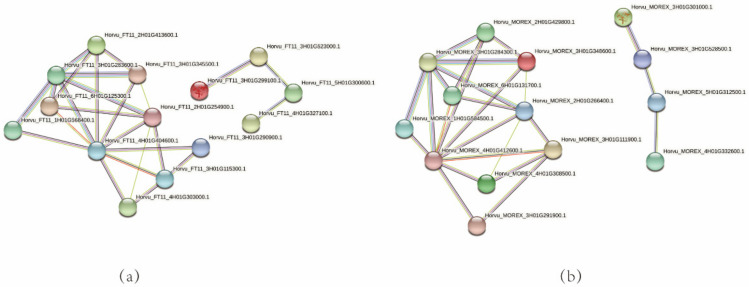
Network interaction analysis in B1K-04-12 (**a**) and Morex (**b**).

**Figure 8 genes-13-00834-f008:**
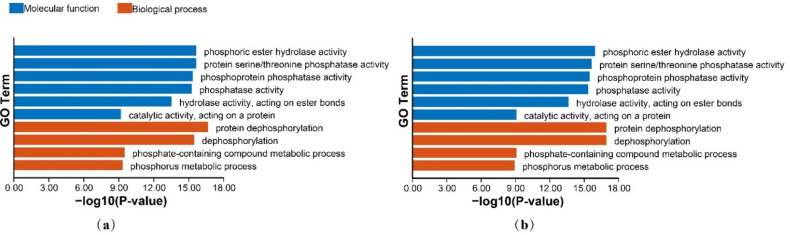
GO enrichment in B1K-04-12 (**a**) and Morex (**b**).

**Figure 9 genes-13-00834-f009:**
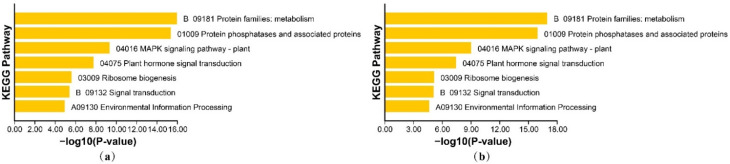
KEGG enrichment in B1K-04-12 (**a**) and Morex (**b**).

**Figure 10 genes-13-00834-f010:**
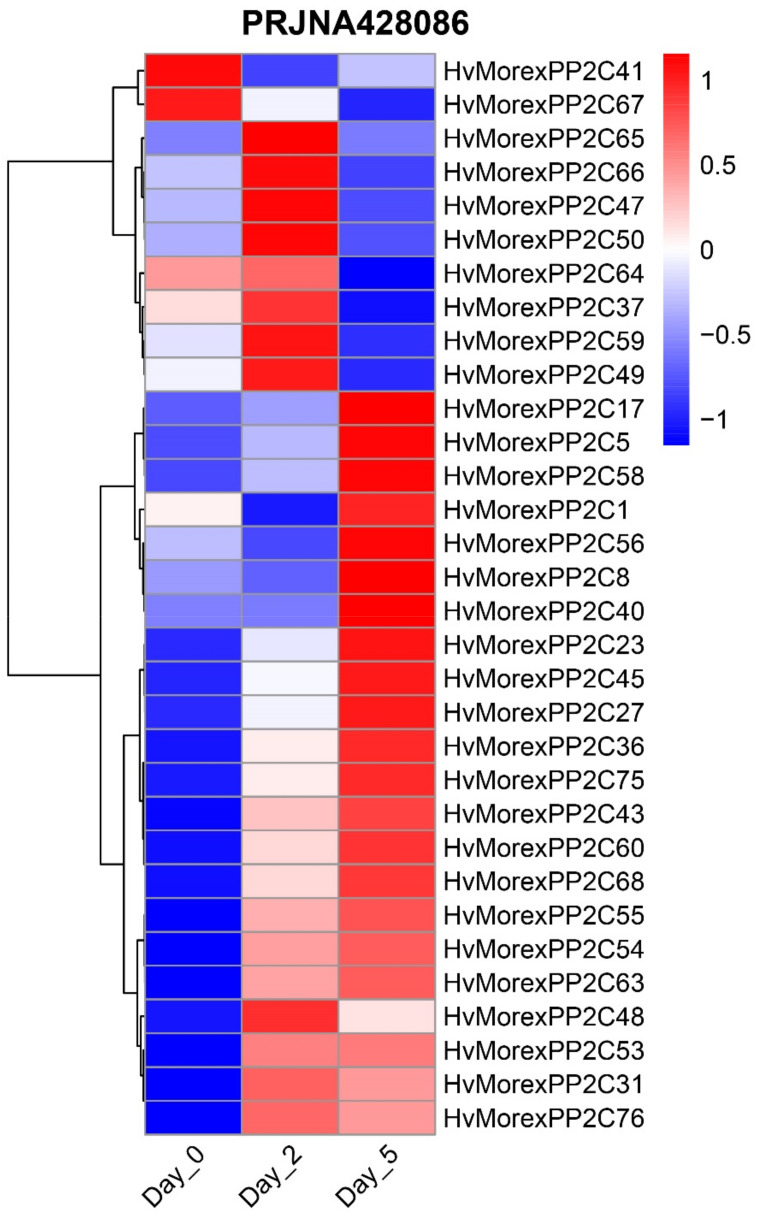
Expression profile of barley PP2C genes during the early stages of barley development.

**Figure 11 genes-13-00834-f011:**
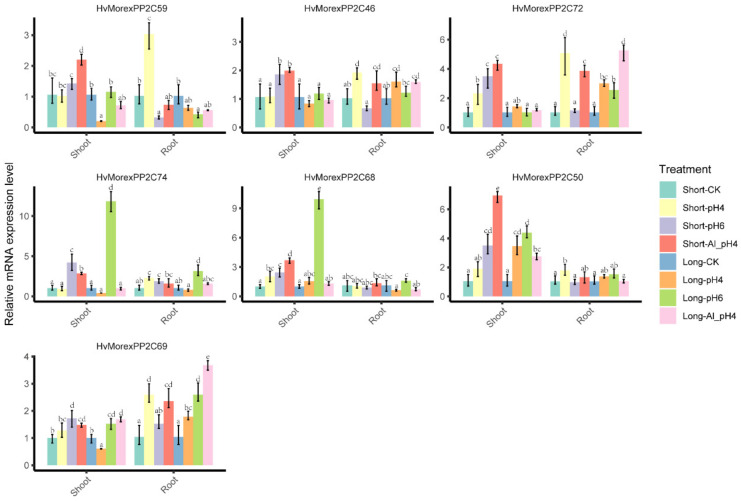
Relative expression levels of seven candidate *HvMorexPP2C* genes after aluminum and low pH stresses in shoot and root, respectively. The values are given as the means ± SDs of three biological replicates. Additionally, the letters show the multiple comparison results of Waller–Duncan method calculated by SPSS.

**Table 1 genes-13-00834-t001:** The number of HvPP2C genes in each accession of barley pan-genome.

Accessions	Number of *PP2Cs*
Hockett	88
Morex	85
HOR_3081	84
HOR_9043	84
HOR_7552	83
OUN333	83
HOR_10350	82
HOR_21599	82
ZDM02064	82
HOR_13821	81
HOR_13942	81
HOR_8148	81
RGT_Planet	81
ZDM01467	81
Akashinriki	80
Barke	80
Golden_Promise	80
HOR_3365	80
Igri	79
B1K-04-12	78

**Table 2 genes-13-00834-t002:** Summary of subcellular localization of *HvPP2Cs* in B1K-04-12 and Morex.

Subcellular Location	B1K-04-12	Morex
Chloroplast	35	38
Cytosol	19	20
Nuclear	16	18
Cytoskeleton	3	3
Peroxisome	2	2
Chloroplast_Mitochondrion	1	1
Endoplasmic reticulum	1	1
Vacuolar membrane	1	2

**Table 3 genes-13-00834-t003:** Ka/Ks of syntenic gene pairs between B1K-04-12 and Morex.

Seq_1	Seq_2	Ka	Ks	Ka_Ks
Horvu_FT11_4H01G404600.1	Horvu_MOREX_4H01G412600.1	0.0034	0.0025	1.3583
Horvu_FT11_2H01G014500.1	Horvu_MOREX_2H01G002500.1	0.0032	0.0060	0.5319
Horvu_FT11_2H01G454900.1	Horvu_MOREX_2H01G471900.1	0.0030	0.0061	0.4947
Horvu_FT11_2H01G088000.1	Horvu_MOREX_2H01G089800.1	0.0023	0.0062	0.3771
Horvu_FT11_6H01G125300.1	Horvu_MOREX_6H01G131700.1	0.0012	0.0033	0.3505
Horvu_FT11_4H01G099700.1	Horvu_MOREX_4H01G099600.1	0.0033	0.0095	0.3460
Horvu_FT11_5H01G121400.1	Horvu_MOREX_5H01G126800.1	0.0012	0.0036	0.3204
Horvu_FT11_3H01G115300.1	Horvu_MOREX_3H01G111900.1	0.0055	0.0172	0.3184
Horvu_FT11_7H01G128600.1	Horvu_MOREX_7H01G158800.1	0.0012	0.0040	0.3146
Horvu_FT11_3H01G283600.1	Horvu_MOREX_3H01G284300.1	0.0007	0.0021	0.3105
Horvu_FT11_6H01G259200.1	Horvu_MOREX_6H01G264100.1	0.0004	0.0013	0.3073
Horvu_FT11_4H01G287000.1	Horvu_MOREX_4H01G292800.1	0.0011	0.0036	0.2993
Horvu_FT11_7H01G042900.1	Horvu_MOREX_7H01G041800.1	0.0071	0.0324	0.2184
Horvu_FT11_7H01G478100.1	Horvu_MOREX_7H01G506400.1	0.0014	0.0080	0.1773
Horvu_FT11_7H01G430700.1	Horvu_MOREX_7H01G456500.1	0.0013	0.0076	0.1718
Horvu_FT11_5H01G113600.1	Horvu_MOREX_5H01G118100.1	0.0012	0.0071	0.1639
Horvu_FT11_2H01G621300.1	Horvu_MOREX_2H01G644100.1	0.0011	0.0070	0.1634
Horvu_FT11_5H01G380300.1	Horvu_MOREX_5H01G396300.1	0.0010	0.0063	0.1626
Horvu_FT11_3H01G711500.1	Horvu_MOREX_3H01G723700.1	0.0011	0.0071	0.1616
Horvu_FT11_1H01G521300.1	Horvu_MOREX_1H01G534000.1	0.0009	0.0078	0.1154
Horvu_FT11_4H01G533500.1	Horvu_MOREX_4H01G549600.1	0.0045	0.0403	0.1110
Horvu_FT11_1H01G107800.1	Horvu_MOREX_1H01G108200.1	0.0012	0.0107	0.1090
Horvu_FT11_1H01G039300.1	Horvu_MOREX_1H01G036200.1	0.0031	0.0387	0.0800
Horvu_FT11_5H01G653000.1	Horvu_MOREX_5H01G674700.1	0.0007	0.0103	0.0683
Horvu_FT11_5H01G628500.1	Horvu_MOREX_5H01G651500.1	0.0019	0.0627	0.0302

## Data Availability

Not applicable.

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
