# Peer review of "Genome-Wide Identification and Transcriptional Expression Profiles of *PP2C* in the Barley (*Hordeum vulgare* L.) Pan-Genome"

_genes, 2022, doi:10.3390/genes13050834_

Round 1
Reviewer 1 Report
Page 2
In the paragraph starts with “It is generally accepted that barley domestication is a single event that happened in the Fertile Crescent [8]. Nevertheless, some other researchers consider Tibet or the Near East as a secondary origin center” Change “a secondary origin center” into “a secondary center of origin”
Page 3
In the paragraph “2.2. Prediction of Protein Physical and Chemical Parameters and Subcellular Localization
The physical and chemical parameters of identified protein sequences were calculated using the Expasy ProtParam tool [33], including molecular weight (MW), theoretical isoelectric point (pI and amino acid composition. WoLF PSORT online service [34] pre[1]dicted protein subcellular localization. The bracket is missing change to (pI )
Page 6:
In the paragraph “The summary of the subcellular localization of the two chosen accessions in this study is shown in Table 2. The results showed that the PP2Cs are randomly distributed in the barley cells. Most of the PP2C genes are located in the chloroplast, and some of them are also located in the cytosol and nuclear.” Did the author mean “nucleus”
Author Response
Response to Reviewer 1 Comments
Point 1: Page 2
In the paragraph starts with “It is generally accepted that barley domestication is a single event that happened in the Fertile Crescent [8]. Nevertheless, some other researchers consider Tibet or the Near East as a secondary origin center” Change “a secondary origin center” into “a secondary center of origin”
Response 1: Language description has been modified. “a secondary origin center” has been changed into “a secondary center of origin” in the manuscript.
Point 2: Page 3
In the paragraph “2.2. Prediction of Protein Physical and Chemical Parameters and Subcellular Localization
The physical and chemical parameters of identified protein sequences were calculated using the Expasy ProtParam tool [33], including molecular weight (MW), theoretical isoelectric point (pI and amino acid composition. WoLF PSORT online service [34] pre[1]dicted protein subcellular localization. The bracket is missing change to (pI )
Response 2: The missing closing bracket has been filled in.
Point 3: Page 6:
In the paragraph “The summary of the subcellular localization of the two chosen accessions in this study is shown in Table 2. The results showed that the PP2Cs are randomly distributed in the barley cells. Most of the PP2C genes are located in the chloroplast, and some of them are also located in the cytosol and nuclear.” Did the author mean “nucleus”
Response 3: Wording mistake has been corrected. “nuclear” has been changed into “nucleus” in the manuscript.
Reviewer 2 Report
Authors compared the available transcriptomic studies of the wild and cultivated barley, related to plant protein phosphatases PP2C. Authors studied also gene structure, conserved motif distribution, duplication events, cis-elements in wild and cultivated barley. Authors identified positively selected homologous genes between wild and cultivated barley that are important for barley domestication. In my opinion research is important, original, well planned and performed. Some minor comments should be addressed:
Section 2.2
Before pI- remove the bracket
Section 2.11
There should be μL not L as a volume unit
Add Celsius degree sign; should be ⁰C not C
How the quality of RNA was assessed
Amount of RNA taken for analysis
Fig 2b and 3b; there is a problem with the picture visibility and clarity. Rectangles that mark particular motifs are very small, moreover colors are sometimes very similar; for example color that mark Motif 7 and PLN03145, Motif and NB-ARC, Motif 10 and PHA03307, Motif 5 and PLN03200, Motif 9 and LRR. Assuming that it could be hard to find more contrast-enough colors, Authors could putatively precisely distinguish these domains using not only rectangles, but circles or triangles on Fig 2b and 3b.
Section 3.10
Why and how authors selected presented 7 genes to test the response to low pH and aluminum stress ?
Discussion section should contain more information related to the response to low pH or aluminum for genes that are homologous to those used in the presented RT-PCR study.
Latin names of organisms should be in italics-check the whole text.
A typhographic errors are present in reference nr 37- addition of sup, use of ss etc.
There is a problem with the access to supplement files.
Author Response
Response to Reviewer 2 Comments
Point 1: Section 2.2
Before pI- remove the bracket
Response 1: The missing closing bracket has been filled in, form a pair with the first half of the bracket.
Point 2: Section 2.11
There should be μL not L as a volume unit
Add Celsius degree sign; should be ⁰C not C
How the quality of RNA was assessed
Amount of RNA taken for analysis
Response 2:
- The volume unit has been changed to μL.
- The Celsius degree sign has been changed to ⁰C.
- The assessment of the quality of RNA was added to the method in Section 2.11. “The quality of total RNA was detected by 1% agarose gel electrophoresis. The concentra-tion of RNA was detected by the NanoDrop-2000 UV spectrophotometer. The samples that had an A260/A280 between 1.8 and 2.0 were used for reverse transcription. “
- 1 µg of total RNA was taken for analysis. The following context was added to the method in Section 2.11. “The reverse transcription assay was based on a 20µL reaction volume containing 1 µg of total RNA.”
Point 3:
Fig 2b and 3b; there is a problem with the picture visibility and clarity. Rectangles that mark particular motifs are very small, moreover colors are sometimes very similar; for example color that mark Motif 7 and PLN03145, Motif and NB-ARC, Motif 10 and PHA03307, Motif 5 and PLN03200, Motif 9 and LRR. Assuming that it could be hard to find more contrast-enough colors, Authors could putatively precisely distinguish these domains using not only rectangles, but circles or triangles on Fig 2b and 3b.
Response 3: The similar color used in Figure 2b and 2c and in Figure 3b and 3c was initially distinguished by the black stroke. The rectangles, which represent motifs in Figure 2b and 3b, had no black strokes. And the rectangles that represent domains had a 0.9 pt black stroke in Figure 2c and 3c. The problem that we cannot use circles or triangles in these figures is that triangles can not represent the length of the motifs or domains accurately; and though circles can represent the length, they need to expand the height and there is not adequate space for more than 80 circles arranged vertically in these figures. Therefore, the width of the black stroke was changed from 0.9 pt to 1.8 pt in Figure 2c and 3c to distinguish them from Figure 2b and 3b. The figures were re-uploaded in the manuscript.
Point 4: Section 3.10
Why and how authors selected presented 7 genes to test the response to low pH and aluminum stress ?
Response 4: Summarizing the results of all 13 RNA-seq analyses, the most abundant gene ex-pression differences under experimental treatments were in aluminum and low pH, while many HvPP2Cs also showed tissue specific expression. Then, 7 genes were randomly selected for qRT-PCR experiments from different expressed genes in aluminum and low pH experiment treatments or with tissue-specific expression summarized in other experiments.
Point 5: Section 3.10
Discussion section should contain more information related to the response to low pH or aluminum for genes that are homologous to those used in the presented RT-PCR study.
Response 5: The following context was added to discussion section.
“Among 7 genes used for the qRT-PCR, 3 of them had a meaningful Ka/Ks value. It includes two purified selection genes, HvMorexPP2C50 and HvMorexPP2C72, with Ka/Ks values of 0.1718 and 0.3184, respectively, and one positive selection gene, HvMorexPP2C74, with Ka/Ks values of 1.3583.”
“qRT-PCR confirmed this positive selection gene HvMorexPP2C74, showing the upregulated expression of pH = 6.0 in both shoot and root after short- or long-term treatments. And it also showed the upregulated expression of pH = 4.0 with Al3+ ions in shoot after short-term treatment. This indicates that this gene might be involved in stress responses in barley.”
Point 6: Latin names of organisms should be in italics-check the whole text.
Response 6: The Latin names of organisms' gene IDs were checked and changed to italics in the whole text.
Point 7: A typhographic errors are present in reference nr 37- addition of sup, use of ss etc.
Response 7: There was a format mistake of reference nr 37 during EndNote insertion. It's now fixed in the manuscript.
Point 8: There is a problem with the access to supplement files.
Response 8: I will re-upload the supplement files into submition system.